# From Bandits to Experts:
# A Tale of Domination and Independence

**Noga Alon**
Tel-Aviv University, Israel
nogaa@tau.ac.il

**Nicolò Cesa-Bianchi**
Università degli Studi di Milano, Italy
nicolo.cesa-bianchi@unimi.it

**Claudio Gentile**
University of Insubria, Italy
claudio.gentile@uninsubria.it

**Yishay Mansour**
Tel-Aviv University, Israel
mansour@tau.ac.il

## Abstract

We consider the partial observability model for multi-armed bandits, introduced by Mannor and Shamir [14]. Our main result is a characterization of regret in the directed observability model in terms of the dominating and independence numbers of the observability graph (which must be accessible before selecting an action). In the undirected case, we show that the learner can achieve optimal regret without even accessing the observability graph before selecting an action. Both results are shown using variants of the Exp3 algorithm operating on the observability graph in a time-efficient manner.

## 1 Introduction

Prediction with expert advice —see, e.g., [13, 16, 6, 10, 7]— is a general abstract framework for studying sequential prediction problems, formulated as repeated games between a player and an adversary. A well studied example of prediction game is the following: In each round, the adversary privately assigns a loss value to each action in a fixed set. Then the player chooses an action (possibly using randomization) and incurs the corresponding loss. The goal of the player is to control regret, which is defined as the excess loss incurred by the player as compared to the best fixed action over a sequence of rounds. Two important variants of this game have been studied in the past: the expert setting, where at the end of each round the player observes the loss assigned to each action for that round, and the bandit setting, where the player only observes the loss of the chosen action, but not that of other actions.

Let $K$ be the number of available actions, and $T$ be the number of prediction rounds. The best possible regret for the expert setting is of order $\sqrt{T \log K}$. This optimal rate is achieved by the Hedge algorithm [10] or the Follow the Perturbed Leader algorithm [12]. In the bandit setting, the optimal regret is of order $\sqrt{TK}$, achieved by the INF algorithm [2]. A bandit variant of Hedge, called Exp3 [3], achieves a regret with a slightly worse bound of order $\sqrt{TK \log K}$.

Recently, Mannor and Shamir [14] introduced an elegant way for defining intermediate observability models between the expert setting (full observability) and the bandit setting (single observability). An intuitive way of representing an observability model is through a directed graph over actions: an arc[1] from action $i$ to action $j$ implies that when playing action $i$ we get information also about the loss of action $j$. Thus, the expert setting is obtained by choosing a complete graph over actions (playing any action reveals all losses), and the bandit setting is obtained by choosing an empty edge set (playing an action only reveals the loss of that action).

The main result of [14] concerns undirected observability graphs. The regret is characterized in terms of the independence number $\alpha$ of the undirected observability graph. Specifically, they prove that $\sqrt{T\alpha \log K}$ is the optimal regret (up to logarithmic factors) and show that a variant of Exp3, called ELP, achieves this bound when the graph is known ahead of time, where $\alpha \in \{1, \ldots, K\}$ interpolates between full observability ($\alpha = 1$ for the clique) and single observability ($\alpha = K$ for the graph with no edges). Given the observability graph, ELP runs a linear program to compute the desired distribution over actions. In the case when the graph changes over time, and at each time step ELP observes the current observability graph before prediction, a bound of $\sqrt{\sum_{t=1}^{T} \alpha_t \log K}$ is shown, where $\alpha_t$ is the independence number of the graph at time $t$. A major problem left open in [14] was the characterization of regret for directed observability graphs, a setting for which they only proved partial results.

Our main result is a full characterization (to within logarithmic factors) of regret in the case of directed observability graphs. Our upper bounds are proven using a new algorithm, called Exp3-DOM. This algorithm is efficient to run even when the graph changes over time: it just needs to compute a small dominating set of the current observability graph (which must be given as side information) before prediction.[2] As in the undirected case, the regret for the directed case is characterized in terms of the independence numbers of the observability graphs (computed ignoring edge directions). We arrive at this result by showing that a key quantity emerging in the analysis of Exp3-DOM can be bounded in terms of the independence numbers of the graphs. This bound (Lemma 13 in the appendix) is based on a combinatorial construction which might be of independent interest.

We also explore the possibility of the learning algorithm receiving the observability graph only after prediction, and not before. For this setting, we introduce a new variant of Exp3, called Exp3-SET, which achieves the same regret as ELP for undirected graphs, but without the need of accessing the current observability graph before each prediction. We show that in some random directed graph models Exp3-SET has also a good performance. In general, we can upper bound the regret of Exp3-SET as a function of the maximum acyclic subgraph of the observability graph, but this upper bound may not be tight. Yet, Exp3-SET is much simpler and computationally less demanding than ELP, which needs to solve a linear program in each round.

There are a variety of real-world settings where partial observability models corresponding to directed and undirected graphs are applicable. One of them is route selection. We are given a graph of possible routes connecting cities: when we select a route $r$ connecting two cities, we observe the cost (say, driving time or fuel consumption) of the "edges" along that route and, in addition, we have complete information on any sub-route $r'$ of $r$, but not vice versa. We abstract this in our model by having an observability graph over routes $r$, and an arc from $r$ to any of its sub-routes $r'$.[3]

Sequential prediction problems with partial observability models also arise in the context of recommendation systems. For example, an online retailer, which advertises products to users, knows that users buying certain products are often interested in a set of related products. This knowledge can be represented as a graph over the set of products, where two products are joined by an edge if and only if users who buy any one of the two are likely to buy the other as well. In certain cases, however, edges have a preferred orientation. For instance, a person buying a video game console might also buy a high-def cable to connect it to the TV set. Vice versa, interest in high-def cables need not indicate an interest in game consoles.

Such observability models may also arise in the case when a recommendation system operates in a network of users. For example, consider the problem of recommending a sequence of products, or contents, to users in a group. Suppose the recommendation system is hosted on an online social network, on which users can befriend each other. In this case, it has been observed that social relationships reveal similarities in tastes and interests [15]. However, social links can also be asymmetric (e.g., followers of celebrities). In such cases, followers might be more likely to shape their preferences after the person they follow, than the other way around. Hence, a product liked by a celebrity is probably also liked by his/her followers, whereas a preference expressed by a follower is more often specific to that person.

___________________________

[2] Computing an approximately minimum dominating set can be done by running a standard greedy set cover algorithm, see Section 2.

[3] Though this example may also be viewed as an instance of combinatorial bandits [8], the model studied here is more general. For example, it does not assume linear losses, which could arise in the routing example from the partial ordering of sub-routes.

## 2 Learning protocol, notation, and preliminaries

As stated in the introduction, we consider an adversarial multi-armed bandit setting with a finite action set $V = \{1, \ldots, K\}$. At each time $t = 1, 2, \ldots$, a player (the "learning algorithm") picks some action $I_t \in V$ and incurs a bounded loss $\ell_{I_t,t} \in [0, 1]$. Unlike the standard adversarial bandit problem [3, 7], where only the played action $I_t$ reveals its loss $\ell_{I_t,t}$, here we assume all the losses in a subset $S_{I_t,t} \subseteq V$ of actions are revealed after $I_t$ is played. More formally, the player observes the pairs $(i, \ell_{i,t})$ for each $i \in S_{I_t,t}$. We also assume $i \in S_{i,t}$ for any $i$ and $t$, that is, any action reveals its own loss when played. Note that the bandit setting ($S_{i,t} = \{i\}$) and the expert setting ($S_{i,t} = V$) are both special cases of this framework. We call $S_{i,t}$ the *observation set* of action $i$ at time $t$, and write $i \xrightarrow{t} j$ when at time $t$ playing action $i$ also reveals the loss of action $j$. Hence, $S_{i,t} = \{j \in V : i \xrightarrow{t} j\}$. The family of observation sets $\{S_{i,t}\}_{i \in V}$ we collectively call the *observation system* at time $t$.

The adversaries we consider are nonoblivious. Namely, each loss $\ell_{i,t}$ at time $t$ can be an arbitrary function of the past player's actions $I_1, \ldots, I_{t-1}$. The performance of a player $A$ is measured through the regret

$$\max_{k \in V} \mathbb{E}[L_{A,T} - L_{k,T}]$$

where $L_{A,T} = \ell_{I_1,1} + \cdots + \ell_{I_T,T}$ and $L_{k,T} = \ell_{k,1} + \cdots + \ell_{k,T}$ are the cumulative losses of the player and of action $k$, respectively. The expectation is taken with respect to the player's internal randomization (since losses are allowed to depend on the player's past random actions, also $L_{k,t}$ may be random).[4] The observation system $\{S_{i,t}\}_{i \in V}$ is also adversarially generated, and each $S_{i,t}$ can be an arbitrary function of past player's actions, just like losses are. However, in Section 3 we also consider a variant in which the observation system is randomly generated according to a specific stochastic model.

Whereas some algorithms need to know the observation system at the beginning of each step $t$, others need not. From this viewpoint, we consider two online learning settings. In the first setting, called the *informed* setting, the full observation system $\{S_{i,t}\}_{i \in V}$ selected by the adversary is made available to the learner *before* making its choice $I_t$. This is essentially the "side-information" framework first considered in [14]. In the second setting, called the *uninformed setting*, no information whatsoever regarding the time-$t$ observation system is given to the learner prior to prediction. We find it convenient to adopt the same graph-theoretic interpretation of observation systems as in [14]. At each step $t = 1, 2, \ldots$, the observation system $\{S_{i,t}\}_{i \in V}$ defines a directed graph $G_t = (V, D_t)$, where $V$ is the set of actions, and $D_t$ is the set of arcs, i.e., ordered pairs of nodes. For $j \neq i$, arc $(i, j) \in D_t$ if and only if $i \xrightarrow{t} j$ (the self-loops created by $i \xrightarrow{t} i$ are intentionally ignored). Hence, we can equivalently define $\{S_{i,t}\}_{i \in V}$ in terms of $G_t$. Observe that the outdegree $d_i^+$ of any $i \in V$ equals $|S_{i,t}| - 1$. Similarly, the indegree $d_i^-$ of $i$ is the number of action $j \neq i$ such that $i \in S_{j,t}$ (i.e., such that $j \xrightarrow{t} i$). A notable special case of the above is when the observation system is symmetric over time: $j \in S_{i,t}$ if and only if $i \in S_{j,t}$ for all $i, j$ and $t$. In words, playing $i$ at time $t$ reveals the loss of $j$ if and only if playing $j$ at time $t$ reveals the loss of $i$. A symmetric observation system is equivalent to $G_t$ being an undirected graph or, more precisely, to a directed graph having, for every pair of nodes $i, j \in V$, either no arcs or length-two directed cycles. Thus, from the point of view of the symmetry of the observation system, we also distinguish between the *directed* case ($G_t$ is a general directed graph) and the *symmetric* case ($G_t$ is an undirected graph for all $t$).

The analysis of our algorithms depends on certain properties of the sequence of graphs $G_t$. Two graph-theoretic notions playing an important role here are those of *independent sets* and *dominating sets*. Given an undirected graph $G = (V, E)$, an independent set of $G$ is any subset $T \subseteq V$ such that no two $i, j \in T$ are connected by an edge in $E$. An independent set is *maximal* if no proper superset thereof is itself an independent set. The size of a largest (maximal) independent set is the *independence number* of $G$, denoted by $\alpha(G)$. If $G$ is directed, we can still associate with it an independence number: we simply view $G$ as undirected by ignoring arc orientation. If $G = (V, D)$ is a directed graph, then a subset $R \subseteq V$ is a dominating set for $G$ if for all $j \notin R$ there exists some $i \in R$ such that arc $(i, j) \in D$. In our bandit setting, a time-$t$ dominating set $R_t$ is a subset of actions with the property that the loss of any remaining action in round $t$ can be observed by playing

**Algorithm 1:** Exp3-SET algorithm (for the uninformed setting)

---

**Parameter:** $\eta \in [0, 1]$
**Initialize:** $w_{i,1} = 1$ for all $i \in V = \{1, \ldots, K\}$
**For** $t = 1, 2, \ldots$:

> 1. Observation system $\{S_{i,t}\}_{i \in V}$ is generated but not disclosed ;
>
> 2. Set $p_{i,t} = \dfrac{w_{i,t}}{W_{i,t}}$ for each $i \in V$, where $W_t = \sum_{j \in V} w_{j,t}$ ;
>
> 3. Play action $I_t$ drawn according to distribution $p_t = (p_{1,t}, \ldots, p_{K,t})$ ;
>
> 4. Observe pairs $(i, \ell_{i,t})$ for all $i \in S_{I_t,t}$;
>
> 5. Observation system $\{S_{i,t}\}_{i \in V}$ is disclosed ;
>
> 6. For any $i \in V$ set $w_{i,t+1} = w_{i,t} \exp(-\eta \widehat{\ell}_{i,t})$, where
> $$\widehat{\ell}_{i,t} = \frac{\ell_{i,t}}{q_{i,t}} \mathbb{I}\{i \in S_{I_t,t}\} \qquad \text{and} \qquad q_{i,t} = \sum_{j \,:\, j \xrightarrow{t} i} p_{j,t} \,.$$

---

some action in $R_t$. A dominating set is *minimal* if no proper subset thereof is itself a dominating set. The domination number of directed graph $G$, denoted by $\gamma(G)$, is the size of a smallest (minimal) dominating set of $G$.

Computing a minimum dominating set for an arbitrary directed graph $G_t$ is equivalent to solving a minimum set cover problem on the associated observation system $\{S_{i,t}\}_{i \in V}$. Although minimum set cover is NP-hard, the well-known Greedy Set Cover algorithm [9], which repeatedly selects from $\{S_{i,t}\}_{i \in V}$ the set containing the largest number of uncovered elements so far, computes a dominating set $R_t$ such that $|R_t| \leq \gamma(G_t)(1 + \ln K)$.

Finally, we can also lift the notion of independence number of an undirected graph to directed graphs through the notion of *maximum acyclic subgraphs*: Given a directed graph $G = (V, D)$, an acyclic subgraph of $G$ is any graph $G' = (V', D')$ such that $V' \subseteq V$, and $D' = D \cap (V' \times V')$, with no (directed) cycles. We denote by $\text{mas}(G) = |V'|$ the maximum size of such $V'$. Note that when $G$ is undirected (more precisely, as above, when $G$ is a directed graph having for every pair of nodes $i, j \in V$ either no arcs or length-two cycles), then $\text{mas}(G) = \alpha(G)$, otherwise $\text{mas}(G) \geq \alpha(G)$. In particular, when $G$ is itself a directed acyclic graph, then $\text{mas}(G) = |V|$.

## 3 Algorithms without Explicit Exploration: The Uninformed Setting

In this section, we show that a simple variant of the Exp3 algorithm [3] obtains optimal regret (to within logarithmic factors) in the symmetric and uninformed setting. We then show that even the harder adversarial directed setting lends itself to an analysis, though with a weaker regret bound.

Exp3-SET (Algorithm 1) runs Exp3 without mixing with the uniform distribution. Similar to Exp3, Exp3-SET uses loss estimates $\widehat{\ell}_{i,t}$ that divide each observed loss $\ell_{i,t}$ by the probability $q_{i,t}$ of observing it. This probability $q_{i,t}$ is simply the sum of all $p_{j,t}$ such that $j \xrightarrow{t} i$ (the sum includes $p_{i,t}$). Next, we bound the regret of Exp3-SET in terms of the key quantity

$$Q_t = \sum_{i \in V} \frac{p_{i,t}}{q_{i,t}} = \sum_{i \in V} \frac{p_{i,t}}{\sum_{j \,:\, j \xrightarrow{t} i} p_{j,t}} \,. \tag{1}$$

Each term $p_{i,t}/q_{i,t}$ can be viewed as the probability of drawing $i$ from $p_t$ conditioned on the event that $i$ was observed. Similar to [14], a key aspect to our analysis is the ability to deterministically and nonvacuously[5] upper bound $Q_t$ in terms of certain quantities defined on $\{S_{i,t}\}_{i \in V}$. We do so in two ways, either irrespective of how small each $p_{i,t}$ may be (this section) or depending on suitable lower bounds on the probabilities $p_{i,t}$ (Section 4). In fact, forcing lower bounds on $p_{i,t}$ is equivalent to adding exploration terms to the algorithm, which can be done only when knowing $\{S_{i,t}\}_{i \in V}$ before each prediction —an information available only in the informed setting.

The following result is the building block for all subsequent results in the uninformed setting.[6]

**Theorem 1** *The regret of Exp3-SET satisfies*

$$\max_{k \in V} \mathbb{E}\big[L_{A,T} - L_{k,T}\big] \le \frac{\ln K}{\eta} + \frac{\eta}{2} \sum_{t=1}^{T} \mathbb{E}[Q_t] \, .$$

As we said, in the adversarial and symmetric case the observation system at time $t$ can be described by an undirected graph $G_t = (V, E_t)$. This is essentially the problem of [14], which they studied in the easier informed setting, where the same quantity $Q_t$ above arises in the analysis of their ELP algorithm. In their Lemma 3, they show that $Q_t \le \alpha(G_t)$, irrespective of the choice of the probabilities $p_t$. When applied to Exp3-SET, this immediately gives the following result.

**Corollary 2** *In the symmetric setting, the regret of Exp3-SET satisfies*

$$\max_{k \in V} \mathbb{E}\big[L_{A,T} - L_{k,T}\big] \le \frac{\ln K}{\eta} + \frac{\eta}{2} \sum_{t=1}^{T} \mathbb{E}[\alpha(G_t)] \, .$$

*In particular, if for constants $\alpha_1, \ldots, \alpha_T$ we have $\alpha(G_t) \le \alpha_t$, $t = 1, \ldots, T$, then setting $\eta = \sqrt{(2 \ln K) / \sum_{t=1}^{T} \alpha_t}$, gives*

$$\max_{k \in V} \mathbb{E}\big[L_{A,T} - L_{k,T}\big] \le \sqrt{2(\ln K) \sum_{t=1}^{T} \alpha_t} \, .$$

The bounds proven in Corollary 2 are equivalent to those proven in [14] (Theorem 2 therein) for the ELP algorithm. Yet, our analysis is much simpler and, more importantly, our algorithm is simpler and more efficient than ELP, which requires solving a linear program at each step. Moreover, unlike ELP, Exp-SET does not require prior knowledge of the observation system $\{S_{i,t}\}_{i \in V}$ at the beginning of each step.

We now turn to the directed setting. We start by considering a setting in which the observation system is stochastically generated. Then, we turn to the harder adversarial setting.

The Erdős-Renyi model is a standard model for random directed graphs $G = (V, D)$, where we are given a density parameter $r \in [0, 1]$ and, for any pair $i, j \in V$, arc $(i, j) \in D$ with independent probability $r$.[7] We have the following result.

**Corollary 3** *Let $G_t$ be generated according to the Erdős-Renyi model with parameter $r \in [0, 1]$. Then the regret of Exp3-SET satisfies*

$$\max_{k \in V} \mathbb{E}\big[L_{A,T} - L_{k,T}\big] \le \frac{\ln K}{\eta} + \frac{\eta \, T}{2r} \left(1 - (1-r)^K\right) \, .$$

*In the above, the expectations $\mathbb{E}[\cdot]$ are w.r.t. both the algorithm's randomization and the random generation of $G_t$ occurring at each round. In particular, setting $\eta = \sqrt{\frac{2r \ln K}{T(1-(1-r)^K)}}$, gives*

$$\max_{k \in V} \mathbb{E}\big[L_{A,T} - L_{k,T}\big] \le \sqrt{\frac{2(\ln K)T \left(1 - (1-r)^K\right)}{r}} \, .$$

Note that as $r$ ranges in $[0, 1]$ we interpolate between the bandit ($r = 0$)[8] and the expert ($r = 1$) regret bounds.

When the observation system is generated by an adversary, we have the following result.

**Corollary 4** *In the directed setting, the regret of Exp3-SET satisfies*

$$\max_{k \in V} \mathbb{E}\big[L_{A,T} - L_{k,T}\big] \le \frac{\ln K}{\eta} + \frac{\eta}{2} \sum_{t=1}^{T} \mathbb{E}[\mathtt{mas}(G_t)] \, .$$

*In particular, if for constants $m_1, \ldots, m_T$ we have $\mathtt{mas}(G_t) \leq m_t$, $t = 1, \ldots, T$, then setting $\eta = \sqrt{(2 \ln K) / \sum_{t=1}^{T} m_t}$, gives*

$$\max_{k \in V} \mathbb{E}\big[L_{A,T} - L_{k,T}\big] \leq \sqrt{2(\ln K) \sum_{t=1}^{T} m_t} \ .$$

Observe that Corollary 4 is a strict generalization of Corollary 2 because, as we pointed out in Section 2, $\mathtt{mas}(G_t) \geq \alpha(G_t)$, with equality holding when $G_t$ is an undirected graph.

As far as lower bounds are concerned, in the symmetric setting, the authors of [14] derive a lower bound of $\Omega\big(\sqrt{\alpha(G)T}\big)$ in the case when $G_t = G$ for all $t$. We remark that similar to the symmetric setting, we can derive a lower bound of $\Omega\big(\sqrt{\alpha(G)T}\big)$. The simple observation is that given a directed graph $G$, we can define a new graph $G'$ which is made undirected just by reciprocating arcs; namely, if there is an arc $(i, j)$ in $G$ we add arcs $(i, j)$ and $(j, i)$ in $G'$. Note that $\alpha(G) = \alpha(G')$. Since in $G'$ the learner can only receive more information than in $G$, any lower bound on $G$ also applies to $G'$. Therefore we derive the following corollary to the lower bound of [14] (Theorem 4 therein).

**Corollary 5** *Fix a directed graph $G$, and suppose $G_t = G$ for all $t$. Then there exists a (randomized) adversarial strategy such that for any $T = \Omega\big(\alpha(G)^3\big)$ and for any learning strategy, the expected regret of the learner is $\Omega\big(\sqrt{\alpha(G)T}\big)$.*

Moreover, standard results in the theory of Erdős-Renyi graphs, at least in the symmetric case (e.g., [11]), show that, when the density parameter $r$ is constant, the independence number of the resulting graph has an inverse dependence on $r$. This fact, combined with the abovementioned lower bound of [14] gives a lower bound of the form $\sqrt{\frac{T}{r}}$, matching (up to logarithmic factors) the upper bound of Corollary 3.

One may wonder whether a sharper lower bound argument exists which applies to the general directed adversarial setting and involves the larger quantity $\mathtt{mas}(G)$. Unfortunately, the above measure does not seem to be related to the optimal regret: Using Claim 1 in the appendix (see proof of Theorem 3) one can exhibit a sequence of graphs each having a large acyclic subgraph, on which the regret of Exp3-SET is still small.

The lack of a lower bound matching the upper bound provided by Corollary 4 is a good indication that something more sophisticated has to be done in order to upper bound $Q_t$ in (1). This leads us to consider more refined ways of allocating probabilities $p_{i,t}$ to nodes. In the next section, we show an allocation strategy that delivers optimal (to within logarithmic factors) regret bounds using prior knowledge of the graphs $G_t$.

## 4 Algorithms with Explicit Exploration: The Informed Setting

We are still in the general scenario where graphs $G_t$ are adversarially generated and directed, but now $G_t$ is made available before prediction. We start by showing a simple example where our analysis of Exp3-SET inherently fails. This is due to the fact that, when the graph induced by the observation system is directed, the key quantity $Q_t$ defined in (1) cannot be nonvacuously upper bounded independent of the choice of probabilities $p_{i,t}$. A way around it is to introduce a new algorithm, called Exp3-DOM, which controls probabilities $p_{i,t}$ by adding an exploration term to the distribution $p_t$. This exploration term is supported on a dominating set of the current graph $G_t$. For this reason, Exp3-DOM requires prior access to a dominating set $R_t$ at each time step $t$ which, in turn, requires prior knowledge of the entire observation system $\{S_{i,t}\}_{i \in V}$.

As announced, the next result shows that, even for simple directed graphs, there exist distributions $p_t$ on the vertices such that $Q_t$ is linear in the number of nodes while the independence number is $1$.[9] Hence, nontrivial bounds on $Q_t$ can be found only by imposing conditions on distribution $p_t$.

**Algorithm 2:** Exp3-DOM algorithm (for the uninformed setting)

**Input:** Exploration parameters $\gamma^{(b)} \in (0, 1]$ for $b \in \{0, 1, \ldots, \lfloor \log_2 K \rfloor\}$

**Initialization:** $w_{i,1}^{(b)} = 1$ for all $i \in V$ and $b \in \{0, 1, \ldots, \lfloor \log_2 K \rfloor\}$

**For** $t = 1, 2, \ldots$ :

    1. Observation system $\{S_{i,t}\}_{i \in V}$ is generated *and disclosed* ;

    2. Compute a dominating set $R_t \subseteq V$ for $G_t$ associated with $\{S_{i,t}\}_{i \in V}$ ;

    3. Let $b_t$ be such that $|R_t| \in \left[2^{b_t}, 2^{b_t+1} - 1\right]$;

    4. Set $W_t^{(b_t)} = \sum_{i \in V} w_{i,t}^{(b_t)}$;

    5. Set $p_{i,t}^{(b_t)} = \left(1 - \gamma^{(b_t)}\right) \dfrac{w_{i,t}^{(b_t)}}{W_t^{(b_t)}} + \dfrac{\gamma^{(b_t)}}{|R_t|} \mathbb{I}\{i \in R_t\}$;

    6. Play action $I_t$ drawn according to distribution $p_t^{(b_t)} = \left(p_{1,t}^{(b_t)}, \ldots, p_{V,t}^{(b_t)}\right)$ ;

    7. Observe pairs $(i, \ell_{i,t})$ for all $i \in S_{I_t,t}$;

    8. For any $i \in V$ set $w_{i,t+1}^{(b_t)} = w_{i,t}^{(b_t)} \exp\left(-\gamma^{(b_t)} \widehat{\ell}_{i,t}^{(b_t)} / 2^{b_t}\right)$, where

$$\widehat{\ell}_{i,t}^{(b_t)} = \frac{\ell_{i,t}}{q_{i,t}^{(b_t)}} \mathbb{I}\{i \in S_{I_t,t}\} \qquad \text{and} \qquad q_{i,t}^{(b_t)} = \sum_{j \,:\, j \xrightarrow{t} i} p_{j,t}^{(b_t)} \, .$$

---

**Fact 6** *Let $G = (V, D)$ be a total order on $V = \{1, \ldots, K\}$, i.e., such that for all $i \in V$, arc $(j, i) \in D$ for all $j = i+1, \ldots, K$. Let $p = (p_1, \ldots, p_K)$ be a distribution on $V$ such that $p_i = 2^{-i}$, for $i < K$ and $p_k = 2^{-K+1}$. Then*

$$Q = \sum_{i=1}^{K} \frac{p_i}{p_i + \sum_{j \,:\, j \to i} p_j} = \sum_{i=1}^{K} \frac{p_i}{\sum_{j=i}^{K} p_j} = \frac{K+1}{2} \, .$$

We are now ready to introduce and analyze the new algorithm Exp3-DOM for the informed and directed setting. Exp3-DOM (see Algorithm 2) runs $\mathcal{O}(\log K)$ variants of Exp3 indexed by $b = 0, 1, \ldots, \lfloor \log_2 K \rfloor$. At time $t$ the algorithm is given observation system $\{S_{i,t}\}_{i \in V}$, and computes a dominating set $R_t$ of the directed graph $G_t$ induced by $\{S_{i,t}\}_{i \in V}$. Based on the size $|R_t|$ of $R_t$, the algorithm uses instance $b_t = \lfloor \log_2 |R_t| \rfloor$ to pick action $I_t$. We use a superscript $b$ to denote the quantities relevant to the variant of Exp3 indexed by $b$. Similarly to the analysis of Exp3-SET, the key quantities are

$$q_{i,t}^{(b)} = \sum_{j \,:\, i \in S_{j,t}} p_{j,t}^{(b)} = \sum_{j \,:\, j \xrightarrow{t} i} p_{j,t}^{(b)} \qquad \text{and} \qquad Q_t^{(b)} = \sum_{i \in V} \frac{p_{i,t}^{(b)}}{q_{i,t}^{(b)}} \, , \qquad b = 0, 1, \ldots, \lfloor \log_2 K \rfloor \, .$$

Let $T^{(b)} = \left\{t = 1, \ldots, T \,:\, |R_t| \in [2^b, 2^{b+1} - 1]\right\}$. Clearly, the sets $T^{(b)}$ are a partition of the time steps $\{1, \ldots, T\}$, so that $\sum_b |T^{(b)}| = T$. Since the adversary adaptively chooses the dominating sets $R_t$, the sets $T^{(b)}$ are random. This causes a problem in tuning the parameters $\gamma^{(b)}$. For this reason, we do not prove a regret bound for Exp3-DOM, where each instance uses a fixed $\gamma^{(b)}$, but for a slight variant (described in the proof of Theorem 7 —see the appendix) where each $\gamma^{(b)}$ is set through a doubling trick.

**Theorem 7** *In the directed case, the regret of Exp3-DOM satisfies*

$$\max_{k \in V} \mathbb{E}\left[L_{A,T} - L_{k,T}\right] \leq \sum_{b=0}^{\lfloor \log_2 K \rfloor} \left( \frac{2^b \ln K}{\gamma^{(b)}} + \gamma^{(b)} \mathbb{E}\left[ \sum_{t \in T^{(b)}} \left(1 + \frac{Q_t^{(b)}}{2^{b+1}}\right) \right] \right) \, . \tag{2}$$

*Moreover, if we use a doubling trick to choose $\gamma^{(b)}$ for each $b = 0, \ldots, \lfloor \log_2 K \rfloor$, then*

$$\max_{k \in V} \mathbb{E}\big[ L_{A,T} - L_{k,T} \big] = \mathcal{O}\left( (\ln K) \, \mathbb{E}\left[ \sqrt{\sum_{t=1}^{T} \left( 4|R_t| + Q_t^{(b_t)} \right)} \right] + (\ln K)\ln(KT) \right). \quad (3)$$

Importantly, the next result shows how bound (3) of Theorem 7 can be expressed in terms of the sequence $\alpha(G_t)$ of independence numbers of graphs $G_t$ whenever the Greedy Set Cover algorithm [9] (see Section 2) is used to compute the dominating set $R_t$ of the observation system at time $t$.

**Corollary 8** *If Step 2 of Exp3-DOM uses the Greedy Set Cover algorithm to compute the dominating sets $R_t$, then the regret of Exp-DOM with doubling trick satisfies*

$$\max_{k \in V} \mathbb{E}\big[ L_{A,T} - L_{k,T} \big] = \mathcal{O}\left( \ln(K) \sqrt{\ln(KT) \sum_{t=1}^{T} \alpha(G_t)} + \ln(K)\ln(KT) \right)$$

*where, for each $t$, $\alpha(G_t)$ is the independence number of the graph $G_t$ induced by observation system $\{S_{i,t}\}_{i \in V}$.*

Comparing Corollary 8 to Corollary 5 delivers the announced characerization in the general adversarial and directed setting. Moreover, a quick comparison between Corollary 2 and Corollary 8 reveals that a symmetric observation system overcomes the advantage of working in an informed setting: The bound we obtained for the uninformed symmetric setting (Corollary 2) is sharper by logarithmic factors than the one we derived for the informed —but more general, i.e., directed— setting (Corollary 8).

## 5 Conclusions and work in progress

We have investigated online prediction problems in partial information regimes that interpolate between the classical bandit and expert settings. We have shown a number of results characterizing prediction performance in terms of: the structure of the observation system, the amount of information available before prediction, the nature (adversarial or fully random) of the process generating the observation system. Our results are substantial improvements over the paper [14] that initiated this interesting line of research. Our improvements are diverse, and range from considering both informed and uninformed settings to delivering more refined graph-theoretic characterizations, from providing more efficient algorithmic solutions to relying on simpler (and often more general) analytical tools.

Some research directions we are currently pursuing are the following: (1) We are currently investigating the extent to which our results could be applied to the case when the observation system $\{S_{i,t}\}_{i \in V}$ may depend on the loss $\ell_{I_t,t}$ of player's action $I_t$. Note that this would prevent a direct construction of an unbiased estimator for unobserved losses, which many worst-case bandit algorithms (including ours —see the appendix) hinge upon. (2) The upper bound contained in Corollary 4 and expressed in terms of $\mathtt{mas}(\cdot)$ is almost certainly suboptimal, even in the uninformed setting, and we are trying to see if more adequate graph complexity measures can be used instead. (3) Our lower bound in Corollary 5 heavily relies on the corresponding lower bound in [14] which, in turn, refers to a constant graph sequence. We would like to provide a more complete characterization applying to sequences of adversarially-generated graphs $G_1, G_2, \ldots, G_T$ in terms of sequences of their corresponding independence numbers $\alpha(G_1), \alpha(G_2), \ldots, \alpha(G_T)$ (or variants thereof), in both the uninformed and the informed settings. (4) All our upper bounds rely on parameters to be tuned as a function of sequences of observation system quantities (e.g., the sequence of independence numbers). We are trying to see if an adaptive learning rate strategy à la [4], based on the observable quantities $Q_t$, could give similar results without such a prior knowledge.

### Acknowledgments

NA was supported in part by an ERC advanced grant, by a USA-Israeli BSF grant, and by the Israeli I-CORE program. NCB acknowledges partial support by MIUR (project ARS TechnoMedia, PRIN 2010-2011, grant no. 2010N5K7EB_003). YM was supported in part by a grant from the Israel Science Foundation, a grant from the United States-Israel Binational Science Foundation (BSF), a grant by Israel Ministry of Science and Technology and the Israeli Centers of Research Excellence (I-CORE) program (Center No. 4/11).

## Footnotes

[1] According to the standard terminology in directed graph theory, throughout this paper a directed edge will be called an *arc*.

[4] Although we defined the problem in terms of losses, our analysis can be applied to the case when actions return rewards $g_{i,t} \in [0, 1]$ via the transformation $\ell_{i,t} = 1 - g_{i,t}$.

[5] An obvious upper bound on $Q_t$ is $K$.

[6] All proofs are given in the supplementary material to this paper.

[7] Self loops, i.e., arcs $(i, i)$ are included by default here.

[8] Observe that $\lim_{r \to 0+} \frac{1-(1-r)^K}{r} = K$.

[9] In this specific example, the maximum acyclic subgraph has size $K$, which confirms the looseness of Corollary 4.

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
