[Supplementary Material · revealNips.pdf]

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

[10] Notice that $|D_s|$ is at least as large as the number of edges of the undirected version of $G_s$ which the independence number $\alpha_s$ actually refers to.

[11] The zero value won't be of our concern here, because if $p_i = 0$, the corresponding term in (7) can be disregarded.

[12] The pseudo-code for the variant of Exp3-DOM using such a doubling trick is not displayed in this extended abstract.

[13] Notice that $\sum_s \overline{Q}_s^{(b)}$ is an observable quantity.

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

# A  Technical lemmas and proofs

This section contains the proofs of all technical results occurring in the main text, along with ancillary graph-theoretic lemmas. Throughout this appendix, $\mathbb{E}_t[\cdot]$ is a shorthand for $\mathbb{E}\big[\cdot \mid I_1, \ldots, I_{t-1}\big]$.

**Proof of Theorem 1**

Following the proof of Exp3 [3], we have

$$
\begin{aligned}
\frac{W_{t+1}}{W_t} &= \sum_{i \in V} \frac{w_{i,t+1}}{W_t} \\
&= \sum_{i \in V} \frac{w_{i,t} \, \exp(-\eta \, \widehat{\ell}_{i,t})}{W_t} \\
&= \sum_{i \in V} p_{i,t} \, \exp(-\eta \, \widehat{\ell}_{i,t}) \\
&\leq \sum_{i \in V} p_{i,t} \left( 1 - \eta \widehat{\ell}_{i,t} + \frac{1}{2} \eta^2 (\widehat{\ell}_{i,t})^2 \right) \quad \text{using } e^{-x} \leq 1 - x + x^2/2 \text{ for all } x \geq 0 \\
&\leq 1 - \eta \sum_{i \in V} p_{i,t} \widehat{\ell}_{i,t} + \frac{\eta^2}{2} \sum_{i \in V} p_{i,t} (\widehat{\ell}_{i,t})^2 \;.
\end{aligned}
$$

Taking logs, using $\ln(1-x) \leq -x$ for all $x \geq 0$, and summing over $t = 1, \ldots, T$ yields

$$
\ln \frac{W_{T+1}}{W_1} \leq -\eta \sum_{t=1}^{T} \sum_{i \in V} p_{i,t} \widehat{\ell}_{i,t} + \frac{\eta^2}{2} \sum_{t=1}^{T} \sum_{i \in V} p_{i,t} (\widehat{\ell}_{i,t})^2 \;.
$$

Moreover, for any fixed comparison action $k$, we also have

$$
\ln \frac{W_{T+1}}{W_1} \geq \ln \frac{w_{k,T+1}}{W_1} = -\eta \sum_{t=1}^{T} \widehat{\ell}_{k,t} - \ln K \;.
$$

Putting together and rearranging gives

$$
\sum_{t=1}^{T} \sum_{i \in V} p_{i,t} \widehat{\ell}_{i,t} \leq \sum_{t=1}^{T} \widehat{\ell}_{k,t} + \frac{\ln K}{\eta} + \frac{\eta}{2} \sum_{t=1}^{T} \sum_{i \in V} p_{i,t} (\widehat{\ell}_{i,t})^2 \;. \tag{4}
$$

Note that, for all $i \in V$,

$$
\mathbb{E}_t[\widehat{\ell}_{i,t}] = \sum_{j \,:\, i \in S_{j,t}} p_{j,t} \frac{\ell_{i,t}}{q_{i,t}} = \sum_{j \,:\, j \xrightarrow{t} i} p_{j,t} \frac{\ell_{i,t}}{q_{i,t}} = \frac{\ell_{i,t}}{q_{i,t}} \sum_{j \,:\, j \xrightarrow{t} i} p_{j,t} = \ell_{i,t} \;.
$$

Moreover,

$$
\mathbb{E}_t\big[(\widehat{\ell}_{i,t})^2\big] = \sum_{j \,:\, i \in S_{j,t}} p_{j,t} \frac{\ell_{i,t}^2}{q_{i,t}^2} = \frac{\ell_{i,t}^2}{q_{i,t}^2} \sum_{j \,:\, j \xrightarrow{t} i} p_{j,t} \leq \frac{1}{q_{i,t}^2} \sum_{j \,:\, j \xrightarrow{t} i} p_{j,t} = \frac{1}{q_{i,t}} \;.
$$

Hence, taking expectations $\mathbb{E}_t$ on both sides of (4), and recalling the definition of $Q_t$, we can write

$$
\sum_{t=1}^{T} \sum_{i \in V} p_{i,t} \, \ell_{i,t} \leq \sum_{t=1}^{T} \ell_{k,t} + \frac{\ln K}{\eta} + \frac{\eta}{2} \sum_{t=1}^{T} Q_t \;. \tag{5}
$$

Finally, taking expectations to remove conditioning gives

$$
\mathbb{E}\big[L_{A,T} - L_{k,T}\big] \leq \frac{\ln K}{\eta} + \frac{\eta}{2} \sum_{t=1}^{T} \mathbb{E}[Q_t] \;,
$$

as claimed. $\qquad\qquad\qquad\qquad\qquad\qquad\qquad\qquad\qquad\qquad\qquad\qquad\qquad\qquad\qquad\square$

**Proof of Corollary 3**

Fix round $t$, and let $G = (V, D)$ be the Erdős-Renyi random graph generated at time $t$, $N_i^-$ be the in-neighborhood of node $i$, i.e., the set of nodes $j$ such that $(j, i) \in D$, and denote by $d_i^-$ the indegree of $i$.

**Claim 1** *Let $p_1, \ldots, p_K$ be an arbitrary probability distribution defined over $V$, $f : V \to V$ be an arbitrary permutation of $V$, and $\mathbb{E}_f$ denote the expectation w.r.t. permutation $f$ when $f$ is drawn uniformly at random. Then, for any $i \in V$, we have*

$$\mathbb{E}_f \left[ \frac{p_{f(i)}}{p_{f(i)} + \sum_{j\,:\,f(j) \in N^-_{f(i)}} p_{f(j)}} \right] = \frac{1}{1 + d^-_i} \ .$$

**Proof.** Consider selecting a subset $S \subset V$ of $1 + d^-_i$ nodes. We shall consider the contribution to the expectation when $S = N^-_{f(i)} \cup \{f(i)\}$. Since there are $K(K-1) \cdots (K - d^-_i + 1)$ terms (out of $K!$) contributing to the expectation, we can write

$$
\begin{aligned}
\mathbb{E}_f \left[ \frac{p_{f(i)}}{p_{f(i)} + \sum_{j\,:\,f(j) \in N^-_{f(i)}} p_{f(j)}} \right] &= \frac{1}{\binom{K}{d^-_i}} \sum_{S \subset V, |S| = d^-_i} \frac{1}{1 + d^-_i} \sum_{i \in S} \frac{p_i}{p_i + \sum_{j \in S, j \neq i} p_j} \\
&= \frac{1}{\binom{K}{d^-_i}} \sum_{S \subset V, |S| = d^-_i} \frac{1}{1 + d^-_i} \\
&= \frac{1}{1 + d^-_i} \ .
\end{aligned}
$$

$\square$

**Claim 2** *Let $p_1, \ldots, p_K$ be an arbitrary probability distribution defined over $V$, and $\mathbb{E}$ denote the expectation w.r.t. the Erdős-Renyi random draw of arcs at time $t$. Then, for any fixed $i \in V$, we have*

$$\mathbb{E}\left[ \frac{p_i}{p_i + \sum_{j\,:\,j \xrightarrow{t} i} p_j} \right] = \frac{1}{rK} \left( 1 - (1-r)^K \right) \ .$$

**Proof.** For the given $i \in V$ and time $t$, consider the Bernoulli random variables $X_j, j \in V \setminus \{i\}$, and denote by $\mathbb{E}_{j\,:\,j\neq i}$ the expectation w.r.t. all of them. We symmetrize $\mathbb{E}\left[ \frac{p_i}{p_i + \sum_{j\,:\,j \xrightarrow{t} i} p_j} \right]$ by means of a random permutation $f$, as in Claim 1. We can write

$$
\begin{aligned}
\mathbb{E}\left[ \frac{p_i}{p_i + \sum_{j\,:\,j \xrightarrow{t} i} p_j} \right] &= \mathbb{E}_{j\,:\,j\neq i}\left[ \frac{p_i}{p_i + \sum_{j\,:\,j\neq i} X_j p_j} \right] \\
&= \mathbb{E}_{j\,:\,j\neq i} \mathbb{E}_f \left[ \frac{p_{f(i)}}{p_{f(i)} + \sum_{j\,:\,j\neq i} X_{f(j)} p_{f(j)}} \right] \qquad \text{(by symmetry)} \\
&= \mathbb{E}_{j\,:\,j\neq i}\left[ \frac{1}{1 + \sum_{j\,:\,j\neq i} X_j} \right] \qquad \text{(from Claim 1)} \\
&= \sum_{i=0}^{K-1} \binom{K-1}{i} r^i (1-r)^{K-1-i} \frac{1}{i+1} \\
&= \frac{1}{rK} \sum_{i=0}^{K-1} \binom{K}{i+1} r^{i+1} (1-r)^{K-1-i} \\
&= \frac{1}{rK} \left( 1 - (1-r)^K \right) \ .
\end{aligned}
$$

$\square$

At this point, we follow the proof of Theorem 1 up until (5). We take an expectation $\mathbb{E}_{G_1, \ldots, G_T}$ w.r.t. the randomness in generating the sequence of graphs $G_1, \ldots, G_T$. This yields

$$\sum_{t=1}^{T} \mathbb{E}_{G_1, \ldots, G_T} \left[ \sum_{i \in V} p_{i,t}\, \ell_{i,t} \right] \leq \sum_{t=1}^{T} \ell_{k,t} + \frac{\ln K}{\eta} + \frac{\eta}{2} \sum_{t=1}^{T} \mathbb{E}_{G_1, \ldots, G_T} \left[ Q_t \right] \ .$$

We use Claim 2 to upper bound $\mathbb{E}_{G_1,\dots,G_T}[Q_t]$ by $\frac{1}{r}\left(1-(1-r)^K\right)$, and take the outer expectation to remove conditioning, as in the proof of Theorem 1. This concludes the proof. □

The following lemma can be seen as a generalization of Lemma 3 in [14].

**Lemma 9** *Let $G = (V, D)$ be a directed graph with vertex set $V = \{1,\dots,K\}$, and arc set $D$. Let $N_i^-$ be the in-neighborhood of node $i$, i.e., the set of nodes $j$ such that $(j,i) \in D$. Then*

$$\sum_{i=1}^{K} \frac{p_i}{p_i + \sum_{j \in N_i^-} p_j} \leq \mathrm{mas}(G) .$$

**Proof.** We will show that there is a subset of vertices $V'$ such that the induced graph is acyclic and $|V'| \geq \sum_{i=1}^{K} \frac{p_i}{p_i + \sum_{j \in N_i^-} p_j}$.

We prove the lemma by growing set $V'$ starting off from $V' = \emptyset$. Let

$$\Phi_0 = \sum_{i=1}^{K} \frac{p_i}{p_i + \sum_{j \in N_i^-} p_j} ,$$

and $i_1$ be the vertex which minimizes $p_i + \sum_{j \in N_i^-} p_j$ over $i \in V$. We are going to delete $i_1$ from the graph, along with all its incoming neighbors (set $N_{i_1}^-$), and all edges which are incident (both departing and incoming) to these nodes, and then iterating on the remaining graph. Let us denote the in-neighborhoods of the shrunken graph from the first step by $N_{i,1}^-$.

The contribution of all the deleted vertices to $\Phi_0$ is

$$\sum_{r \in N_{i_1}^- \cup \{i_1\}} \frac{p_r}{p_r + \sum_{j \in N_r^-} p_j} \leq \sum_{r \in N_{i_1}^- \cup \{i_1\}} \frac{p_r}{p_{i_1} + \sum_{j \in N_{i_1}^-} p_j} = 1 ,$$

where the inequality follows from the minimality of $i_1$.

Let $V' \leftarrow V' \cup \{i_1\}$, and $V_1 = V - (N_{i_1}^- \cup \{i_1\})$. Then from the first step we have

$$\Phi_1 = \sum_{i \in V_1} \frac{p_i}{p_i + \sum_{j \in N_{i,1}^-} p_j} \geq \sum_{i \in V_1} \frac{p_i}{p_i + \sum_{j \in N_i^-} p_j} \geq \Phi_0 - 1 .$$

We apply the very same argument to $\Phi_1$ with node $i_2$ (minimizing $p_i + \sum_{j \in N_{i,1}^-} p_j$ over $i \in V_1$), to $\Phi_2$ with node $i_3$, ..., to $\Phi_{s-1}$ with node $i_s$, up until $\Phi_s = 0$, i.e., up until no nodes are left in the shrunken graph. This gives $\Phi_0 \leq s = |V'|$, where $V' = \{i_1, i_2, \dots, i_s\}$. Moreover, since in each step $r = 1, \dots, s$ we remore all remaining arcs incoming to $i_r$, the graph induced by set $V'$ cannot contain cycles. □

**Proof of Corollary 4**
The claim follows from a direct combination of Theorem 1 with Lemma 9. □

**Proof of Fact 6**
Using standard properties of geometric sums, one can immediately see that

$$\sum_{i=1}^{K} \frac{p_i}{\sum_{j=i}^{K} p_j} = \sum_{i=1}^{K-1} \frac{2^{-i}}{2^{-i+1}} + \frac{2^{-K+1}}{2^{-K+1}} = \frac{K-1}{2} + 1 = \frac{K+1}{2} ,$$

hence the claimed result. □

The following graph-theoretic lemma turns out to be fairly useful for analyzing directed settings. It is a directed-graph counterpart to a well-known result [5, 17] holding for undirected graphs.

**Lemma 10** *Let $G = (V, D)$ be a directed graph, with $V = \{1,\dots,K\}$. Let $d_i^-$ be the indegree of node $i$, and $\alpha = \alpha(G)$ be the independence number of $G$. Then*

$$\sum_{i=1}^{K} \frac{1}{1 + d_i^-} \leq 2\alpha \ln\left(1 + \frac{K}{\alpha}\right) .$$

**Proof.** We will proceed by induction, starting off from the original $K$-node graph $G = G_K$ with indegrees $\{d_i^-\}_{i=1}^K = \{d_{i,K}^-\}_{i=1}^K$, and independence number $\alpha = \alpha_K$, and then progressively shrink $G$ by eliminating nodes and incident (both departing and incoming) arcs, thereby obtaining a sequence of smaller and smaller graphs $G_K, G_{K-1}, G_{K-2}, \ldots$, and associated indegrees $\{d_{i,K}^-\}_{i=1}^K$, $\{d_{i,K-1}^-\}_{i=1}^{K-1}, \{d_{i,K-2}^-\}_{i=1}^{K-2}, \ldots$, and independence numbers $\alpha_K, \alpha_{K-1}, \alpha_{K-2}, \ldots$. Specifically, in step $s$ we sort nodes $i = 1, \ldots, s$ of $G_s$ in nonincreasing value of $d_{i,s}^-$, and obtain $G_{s-1}$ from $G_s$ by eliminating node 1 (i.e., one having the largest indegree among the nodes of $G_s$), along with its incident arcs. On all such graphs, we will use the classical Turan's theorem (e.g., [1]) stating that any *undirected* graph with $n_s$ nodes and $m_s$ edges has an independent set of size at least $\frac{n_s}{\frac{2m_s}{n_s}+1}$.

This implies that if $G_s = (V_s, D_s)$, then $\alpha_s$ satisfies[10]

$$\frac{|D_s|}{|V_s|} \geq \frac{|V_s|}{2\alpha_s} - \frac{1}{2} . \tag{6}$$

We then start from $G_K$. We can write

$$d_{1,K}^- = \max_{i=1\ldots K} d_{i,K}^- \geq \frac{1}{K} \sum_{i=1}^K d_{i,K}^- = \frac{|D_K|}{|V_K|} \geq \frac{|V_K|}{2\alpha_K} - \frac{1}{2} .$$

Hence,

$$
\begin{aligned}
\sum_{i=1}^K \frac{1}{1 + d_{i,K}^-} &= \frac{1}{1 + d_{1,K}^-} + \sum_{i=2}^K \frac{1}{1 + d_{i,K}^-} \\
&\leq \frac{2\alpha_K}{\alpha_K + K} + \sum_{i=2}^K \frac{1}{1 + d_{i,K}^-} \\
&\leq \frac{2\alpha_K}{\alpha_K + K} + \sum_{i=1}^{K-1} \frac{1}{1 + d_{i,K-1}^-},
\end{aligned}
$$

where the last inequality follows from $d_{i+1,K}^- \geq d_{i,K-1}^-$, $i = 1, \ldots K-1$, due to the arc elimination turning $G_K$ into $G_{K-1}$. Recursively applying the very same argument to $G_{K-1}$ (i.e., to the sum $\sum_{i=1}^{K-1} \frac{1}{1+d_{i,K-1}^-}$), and then iterating all the way to $G_1$ yields the upper bound

$$\sum_{i=1}^K \frac{1}{1 + d_{i,K}^-} \leq \sum_{i=1}^K \frac{2\alpha_i}{\alpha_i + i} .$$

Combining with $\alpha_i \leq \alpha_K = \alpha$, and $\sum_{i=1}^K \frac{1}{\alpha+i} \leq \ln\left(1 + \frac{K}{\alpha}\right)$ concludes the proof. $\qquad\square$

The next lemma relates the size $|R_t|$ of the dominating set $R_t$ computed by the Greedy Set Cover algorithm of [9] operating on the time-$t$ observation system $\{S_{i,t}\}_{i \in V}$ to the independence number $\alpha(G_t)$ and the domination number $\gamma(G_t)$ of $G_t$.

**Lemma 11** *Let $\{S_i\}_{i \in V}$ be an observation system, and $G = (V, D)$ be the induced directed graph, with vertex set $V = \{1, \ldots, K\}$, independence number $\alpha = \alpha(G)$, and domination number $\gamma = \gamma(G)$. Then the dominating set $R$ constructed by the Greedy Set Cover algorithm (see Section 2) satisfies*

$$|R| \leq \min\{\gamma(1 + \ln K), \lceil 2\alpha \ln K \rceil + 1\} .$$

**Proof.** As recalled in Section 2, the Greedy Set Cover algorithm of [9] achieves $|R| \leq \gamma(1 + \ln K)$. In order to prove the other bound, consider the sequence of graphs $G = G_1, G_2, \ldots$, where each $G_{s+1} = (V_{s+1}, D_{s+1})$ is obtained by removing from $G_s$ the vertex $i_s$ selected by the Greedy Set

Cover algorithm, together with all the vertices in $G_s$ that are dominated by $i_s$, and all arcs incident to these vertices. By definition of the algorithm, the outdegree $d_s^+$ of $i_s$ in $G_s$ is largest in $G_s$. Hence,

$$d_s^+ \geq \frac{|D_s|}{|V_s|} \geq \frac{|V_s|}{2\alpha_s} - \frac{1}{2} \geq \frac{|V_s|}{2\alpha} - \frac{1}{2}$$

by Turan's theorem (e.g., [1]), where $\alpha_s$ is the independence number of $G_s$ and $\alpha \geq \alpha_s$. This shows that

$$|V_{s+1}| = |V_s| - d_s^+ - 1 \leq |V_s| \left(1 - \frac{1}{2\alpha}\right) \leq |V_s| e^{-1/(2\alpha)} .$$

Iterating, we obtain $|V_s| \leq K\, e^{-s/(2\alpha)}$. Choosing $s = \lceil 2\alpha \ln K \rceil + 1$ gives $|V_s| < 1$, thereby covering all nodes. Hence the dominating set $R = \{i_1, \ldots, i_s\}$ so constructed satisfies $|R| \leq \lceil 2\alpha \ln K \rceil + 1$. $\qquad\square$

**Lemma 12** *If $a, b \geq 0$, and $a + b \geq B > A > 0$, then*

$$\frac{a}{a + b - A} \leq \frac{a}{a + b} + \frac{A}{B - A} .$$

**Proof.**

$$\frac{a}{a + b - A} - \frac{a}{a + b} = \frac{aA}{(a + b)(a + b - A)} \leq \frac{A}{a + b - A} \leq \frac{A}{B - A} .$$

$\qquad\square$

We now lift Lemma 10 to a more general statement.

**Lemma 13** *Let $G = (V, D)$ be a directed graph, with vertex set $V = \{1, \ldots, K\}$, and arc set $D$. Let $N_i^-$ be the in-neighborhood of node $i$, i.e., the set of nodes $j$ such that $(j, i) \in D$. Let $\alpha$ be the independence number of $G$, $R \subseteq V$ be a dominating set for $G$ of size $r = |R|$, and $p_1, \ldots, p_K$ be a probability distribution defined over $V$, such that $p_i \geq \beta > 0$, for $i \in R$. Then*

$$\sum_{i=1}^{K} \frac{p_i}{p_i + \sum_{j \in N_i^-} p_j} \leq 2\alpha \ln \left(1 + \frac{\lceil \frac{K^2}{r\beta} \rceil + K}{\alpha}\right) + 2r .$$

**Proof.** The idea is to appropriately discretize the probability values $p_i$, and then upper bound the discretized counterpart of $\sum_{i=1}^{K} \frac{p_i}{p_i + \sum_{j \in N_i^-} p_j}$ by reducing to an expression that can be handled by Lemma 10. In order to make this discretization effective, we need to single out the terms $\frac{p_i}{p_i + \sum_{j \in N_i^-} p_j}$ corresponding to nodes $i \in R$. We first write

$$\sum_{i=1}^{K} \frac{p_i}{p_i + \sum_{j \in N_i^-} p_j} = \sum_{i \in R} \frac{p_i}{p_i + \sum_{j \in N_i^-} p_j} + \sum_{i \notin R} \frac{p_i}{p_i + \sum_{j \in N_i^-} p_j}$$
$$\leq r + \sum_{i \notin R} \frac{p_i}{p_i + \sum_{j \in N_i^-} p_j} , \tag{7}$$

and then focus on (7).

Let us discretize the unit interval[11] $(0, 1]$ into subintervals $(\frac{j-1}{M}, \frac{j}{M}]$, $j = 1, \ldots, M$, where $M = \lceil \frac{K^2}{r\beta} \rceil$. Let $\widehat{p}_i = j/M$ be the discretized version of $p_i$, being $j$ the unique integer such that

$$\widehat{p}_i - 1/M < p_i \leq \widehat{p}_i .$$

Let us focus on a single node $i \notin R$ with indegree $d_i^- = |N_i^-|$, and introduce the shorthand notation $P_i = \sum_{j \in N_i^-} p_j$, and $\widehat{P}_i = \sum_{j \in N_i^-} \widehat{p}_j$. We have that $\widehat{P}_i \geq P_i \geq \beta$, since $i$ is dominated by some node $j \in R \cap N_i^-$ such that $p_j \geq \beta$. Moreover, $P_i > \widehat{P}_i - \frac{d_i^-}{M} \geq \beta - \frac{d_i^-}{M} > 0$, and $\widehat{p}_i + \widehat{P}_i \geq \beta$. Hence, for any fixed node $i \notin R$, we can write

$$
\begin{aligned}
\frac{p_i}{p_i + P_i} &\leq \frac{\widehat{p}_i}{\widehat{p}_i + P_i} \\
&< \frac{\widehat{p}_i}{\widehat{p}_i + \widehat{P}_i - \frac{d_i^-}{M}} \\
&\leq \frac{\widehat{p}_i}{\widehat{p}_i + \widehat{P}_i} + \frac{d_i^-/M}{\beta - d_i^-/M} \\
&= \frac{\widehat{p}_i}{\widehat{p}_i + \widehat{P}_i} + \frac{d_i^-}{\beta M - d_i^-} \\
&< \frac{\widehat{p}_i}{\widehat{p}_i + \widehat{P}_i} + \frac{r}{K - r},
\end{aligned}
$$

where in the second-last inequality we used Lemma 12 with $a = \widehat{p}_i$, $b = \widehat{P}_i$, $A = d_i^-/M$, and $B = \beta > d_i^-/M$. Recalling (7), and summing over $i$ then gives

$$
\sum_{i=1}^{K} \frac{p_i}{p_i + P_i} \leq r + \sum_{i \notin R} \frac{\widehat{p}_i}{\widehat{p}_i + \widehat{P}_i} + r = \sum_{i \notin R} \frac{\widehat{p}_i}{\widehat{p}_i + \widehat{P}_i} + 2r. \tag{8}
$$

Therefore, we continue by bounding from above the right-hand side of (8). We first observe that

$$
\sum_{i \notin R} \frac{\widehat{p}_i}{\widehat{p}_i + \widehat{P}_i} = \sum_{i \notin R} \frac{\widehat{s}_i}{\widehat{s}_i + \widehat{S}_i}, \qquad \widehat{S}_i = \sum_{j \in N_i^-} \widehat{s}_j, \tag{9}
$$

where $\widehat{s}_i = M\widehat{p}_i$, $i = 1, \ldots, K$, are integers. Based on the original graph $G$, we construct a new graph $\widehat{G}$ made up of connected cliques. In particular:

- Each node $i$ of $G$ is replaced in $\widehat{G}$ by a clique $C_i$ of size $\widehat{s}_i$; nodes within $C_i$ are connected by length-two cycles.
- If arc $(i, j)$ is in $G$, then for *each* node of $C_i$ draw an arc towards *each* node of $C_j$.

We would like to apply Lemma 10 to $\widehat{G}$. Notice that, by the above construction:

- The independence number of $\widehat{G}$ is the same as that of $G$;
- The indegree $\widehat{d}_k^-$ of each node $k$ in clique $C_i$ satisfies $\widehat{d}_k^- = \widehat{s}_i - 1 + \widehat{S}_i$.
- The total number of nodes of $\widehat{G}$ is

$$
\sum_{i=1}^{K} \widehat{s}_i = M \sum_{i=1}^{K} \widehat{p}_i < M \sum_{i=1}^{K} \left( p_i + \frac{1}{M} \right) = M + K.
$$

Hence, we are in a position to apply Lemma 10 to $\widehat{G}$ with indegrees $\widehat{d}_k^-$, revealing that

$$
\sum_{i \notin R} \frac{\widehat{s}_i}{\widehat{s}_i + \widehat{S}_i} = \sum_{i \notin R} \sum_{k \in C_i} \frac{1}{1 + \widehat{d}_k^-} \leq \sum_{i=1}^{K} \sum_{k \in C_i} \frac{1}{1 + \widehat{d}_k^-} \leq 2\alpha \ln \left( 1 + \frac{M + K}{\alpha} \right).
$$

Putting together as in (8) and (9), and recalling the value of $M$ gives the claimed result. $\square$

**Proof of Theorem 7**
We start to bound the contribution to the overall regret of an instance indexed by $b$. When clear from

the context, we remove the superscript $b$ from $\gamma^{(b)}$, $w_{i,t}^{(b)}$, $p_{i,t}^{(b)}$, and other related quantities. For any $t \in T^{(b)}$ we have

$$
\begin{aligned}
\frac{W_{t+1}}{W_t} &= \sum_{i \in V} \frac{w_{i,t+1}}{W_t} \\
&= \sum_{i \in V} \frac{w_{i,t}}{W_t} \exp\!\big(-(\gamma/2^b)\,\widehat{\ell}_{i,t}\big) \\
&= \sum_{i \in R_t} \frac{p_{i,t} - \gamma/|R_t|}{1 - \gamma} \exp\!\big(-(\gamma/2^b)\,\widehat{\ell}_{i,t}\big) + \sum_{i \notin R_t} \frac{p_{i,t}}{1-\gamma} \exp\!\big(-(\gamma/2^b)\,\widehat{\ell}_{i,t}\big) \\
&\le \sum_{i \in R_t} \frac{p_{i,t} - \gamma/|R_t|}{1-\gamma}\left(1 - \frac{\gamma}{2^b}\widehat{\ell}_{i,t} + \frac{1}{2}\left(\frac{\gamma}{2^b}\widehat{\ell}_{i,t}\right)^2\right) + \sum_{i \notin R_t} \frac{p_{i,t}}{1-\gamma}\left(1 - \frac{\gamma}{2^b}\widehat{\ell}_{i,t} + \frac{1}{2}\left(\frac{\gamma}{2^b}\widehat{\ell}_{i,t}\right)^2\right)
\end{aligned}
$$

(using $e^{-x} \le 1 - x + x^2/2$ for all $x \ge 0$)

$$
\le 1 - \frac{\gamma/2^b}{1-\gamma}\sum_{i \in V} p_{i,t}\widehat{\ell}_{i,t} + \frac{\gamma^2/2^b}{1-\gamma}\sum_{i \in R_t}\frac{\widehat{\ell}_{i,t}}{|R_t|} + \frac{1}{2}\frac{(\gamma/2^b)^2}{1-\gamma}\sum_{i \in V} p_{i,t}\big(\widehat{\ell}_{i,t}\big)^2 \; .
$$

Taking logs, upper bounding, and summing over $t \in T^{(b)}$ yields

$$
\ln \frac{W_{|T^{(b)}|+1}}{W_1} \le -\frac{\gamma/2^b}{1-\gamma}\sum_{t \in T^{(b)}}\sum_{i \in V} p_{i,t}\widehat{\ell}_{i,t} + \frac{\gamma^2/2^b}{1-\gamma}\sum_{t \in T^{(b)}}\sum_{i \in R_t}\frac{\widehat{\ell}_{i,t}}{|R_t|} + \frac{1}{2}\frac{(\gamma/2^b)^2}{1-\gamma}\sum_{t \in T^{(b)}}\sum_{i \in V} p_{i,t}\big(\widehat{\ell}_{i,t}\big)^2 \; .
$$

Moreover, for any fixed comparison action $k$, we also have

$$
\ln \frac{W_{|T^{(b)}|+1}}{W_1} \ge \ln \frac{w_{k,|T^{(b)}|+1}}{W_1} = -\frac{\gamma}{2^b}\sum_{t \in T^{(b)}}\widehat{\ell}_{k,t} - \ln K \; .
$$

Putting together, rearranging, and using $1 - \gamma \le 1$ gives

$$
\sum_{t \in T^{(b)}}\sum_{i \in V} p_{i,t}\widehat{\ell}_{i,t} \le \sum_{t \in T^{(b)}}\widehat{\ell}_{k,t} + \frac{2^b \ln K}{\gamma} + \gamma \sum_{t \in T^{(b)}}\sum_{i \in R_t}\frac{\widehat{\ell}_{i,t}}{|R_t|} + \frac{\gamma}{2^{b+1}}\sum_{t \in T^{(b)}}\sum_{i \in V} p_{i,t}\big(\widehat{\ell}_{i,t}\big)^2 \; .
$$

Reintroducing the notation $\gamma^{(b)}$ and summing over $b = 0, 1, \ldots, \lfloor \log_2 K \rfloor$ gives

$$
\sum_{t=1}^{T}\left(\sum_{i \in V} p_{i,t}^{(b_t)}\widehat{\ell}_{i,t}^{(b_t)} - \widehat{\ell}_{k,t}\right) \le \sum_{b=0}^{\lfloor \log_2 K \rfloor}\frac{2^b \ln K}{\gamma^{(b)}} + \sum_{t=1}^{T}\sum_{i \in R_t}\frac{\gamma^{(b_t)}\widehat{\ell}_{i,t}^{(b_t)}}{|R_t|} + \sum_{t=1}^{T}\frac{\gamma^{(b_t)}}{2^{b_t+1}}\sum_{i \in V} p_{i,t}^{(b_t)}\big(\widehat{\ell}_{i,t}^{(b_t)}\big)^2 \; .
$$

$$(10)$$

Now, similarly to the proof of Theorem 1, we have that, for any $i$ and $t$, $\mathbb{E}_t\big[\widehat{\ell}_{i,t}^{(b_t)}\big] = \ell_{i,t}$ and $\mathbb{E}_t\big[(\widehat{\ell}_{i,t}^{(b_t)})^2\big] \le \frac{1}{q_{i,t}^{(b_t)}}$ . Hence, taking expectations $\mathbb{E}_t$ on both sides of (10) and recalling the definition of $Q_t^{(b)}$ gives

$$
\sum_{t=1}^{T}\left(\sum_{i \in V} p_{i,t}^{(b_t)}\ell_{i,t} - \ell_{k,t}\right) \le \sum_{b=0}^{\lfloor \log_2 K \rfloor}\frac{2^b \ln K}{\gamma^{(b)}} + \sum_{t=1}^{T}\sum_{i \in R_t}\frac{\gamma^{(b_t)}\ell_{i,t}}{|R_t|} + \sum_{t=1}^{T}\frac{\gamma^{(b_t)}}{2^{b_t+1}}Q_t^{(b_t)} \; . \qquad (11)
$$

Moreover,

$$
\sum_{t=1}^{T}\sum_{i \in R_t}\frac{\gamma^{(b_t)}\ell_{i,t}}{|R_t|} \le \sum_{t=1}^{T}\sum_{i \in R_t}\frac{\gamma^{(b_t)}}{|R_t|} = \sum_{t=1}^{T}\gamma^{(b_t)} = \sum_{b=0}^{\lfloor \log_2 K \rfloor}\gamma^{(b)}|T^{(b)}|
$$

and

$$
\sum_{t=1}^{T}\frac{\gamma^{(b_t)}}{2^{b_t+1}}Q_t^{(b_t)} = \sum_{b=0}^{\lfloor \log_2 K \rfloor}\frac{\gamma^{(b)}}{2^{b+1}}\sum_{t \in T^{(b)}} Q_t^{(b)} \; .
$$

Hence, plugging back into (11), taking outer expectations on both sides and recalling that $T^{(b)}$ is random (since the adversary adaptively decides which steps $t$ fall into $T^{(b)}$), we get

$$
\mathbb{E}\big[L_{A,T} - L_{k,T}\big] \leq \sum_{b=0}^{\lfloor \log_2 K \rfloor} \mathbb{E}\left[\frac{2^b \ln K}{\gamma^{(b)}} + \gamma^{(b)}|T^{(b)}| + \frac{\gamma^{(b)}}{2^{b+1}} \sum_{t \in T^{(b)}} Q_t^{(b)}\right]
$$

$$
= \sum_{b=0}^{\lfloor \log_2 K \rfloor} \left(\frac{2^b \ln K}{\gamma^{(b)}} + \gamma^{(b)}\mathbb{E}\left[\sum_{t \in T^{(b)}}\left(1 + \frac{Q_t^{(b)}}{2^{b+1}}\right)\right]\right). \qquad (12)
$$

This establishes (2).

In order to prove inequality (3), we need to tune each $\gamma^{(b)}$ separately. However, a good choice of $\gamma^{(b)}$ depends on the unknown random quantity

$$
\overline{Q}^{(b)} = \sum_{t \in T^{(b)}}\left(1 + \frac{Q_t^{(b)}}{2^{b+1}}\right).
$$

To overcome this problem, we slightly modify Exp3-DOM by applying a doubling trick[12] to guess $\overline{Q}^{(b)}$ for each $b$. Specifically, for each $b = 0, 1, \ldots, \lfloor \log_2 K \rfloor$, we use a sequence $\gamma_r^{(b)} = \sqrt{(2^b \ln K)/2^r}$, for $r = 0, 1, \ldots$. We initially run the algorithm with $\gamma_0^{(b)}$. Whenever the algorithm is running with $\gamma_r^{(b)}$ and observes that $\sum_s \overline{Q}_s^{(b)} > 2^r$, where the sum is over all $s$ so far in $T^{(b)}$,[13] then we restart the algorithm with $\gamma_{r+1}^{(b)}$. Because the contribution of instance $b$ to (12) is

$$
\frac{2^b \ln K}{\gamma^{(b)}} + \gamma^{(b)} \sum_{t \in T^{(b)}}\left(1 + \frac{Q_t^{(b)}}{2^{b+1}}\right),
$$

the regret we pay when using any $\gamma_r^{(b)}$ is at most $2\sqrt{(2^b \ln K)2^r}$. The largest $r$ we need is $\lceil \log_2 \overline{Q}^{(b)} \rceil$ and

$$
\sum_{r=0}^{\lceil \log_2 \overline{Q}^{(b)} \rceil} 2^{r/2} < 5\sqrt{\overline{Q}^{(b)}}.
$$

Since we pay regret at most 1 for each restart, we get

$$
\mathbb{E}\big[L_{A,T} - L_{k,T}\big] \leq c \sum_{b=0}^{\lfloor \log_2 K \rfloor} \mathbb{E}\left[\sqrt{(\ln K)\left(2^b|T^{(b)}| + \frac{1}{2}\sum_{t \in T^{(b)}} Q_t^{(b)}\right)} + \lceil \log_2 \overline{Q}^{(b)} \rceil\right].
$$

for some positive constant $c$. Taking into account that

$$
\sum_{b=0}^{\lfloor \log_2 K \rfloor} 2^b|T^{(b)}| \leq 2\sum_{t=1}^{T} |R_t|
$$

$$
\sum_{b=0}^{\lfloor \log_2 K \rfloor} \sum_{t \in T^{(b)}} Q_t^{(b)} = \sum_{t=1}^{T} Q_t^{(b_t)}
$$

$$
\sum_{b=0}^{\lfloor \log_2 K \rfloor} \lceil \log_2 \overline{Q}^{(b)} \rceil = \mathcal{O}\big((\ln K)\ln(KT)\big),
$$

we obtain

$$\mathbb{E}\big[L_{A,T} - L_{k,T}\big] \leq c \sum_{b=0}^{\lfloor \log_2 K \rfloor} \mathbb{E}\left[\sqrt{(\ln K)\left(2^b |T^{(b)}| + \frac{1}{2}\sum_{t \in T^{(b)}} Q_t^{(b)}\right)}\right] + \mathcal{O}\big((\ln K)\ln(KT)\big)$$

$$\leq c \lfloor \log_2 K \rfloor \mathbb{E}\left[\sqrt{\frac{\ln K}{\lfloor \log_2 K \rfloor}\sum_{t=1}^{T}\left(2|R_t| + \frac{1}{2}Q_t^{(b_t)}\right)}\right] + \mathcal{O}\big((\ln K)\ln(KT)\big)$$

$$= \mathcal{O}\left((\ln K)\,\mathbb{E}\left[\sqrt{\sum_{t=1}^{T}\left(4|R_t| + Q_t^{(b_t)}\right)}\right] + (\ln K)\ln(KT)\right)$$

as desired. $\qquad\square$

**Proof of Corollary 8**

We start off from the upper bound (3) in the statement of Theorem 7. We want to bound the quantities $|R_t|$ and $Q_t^{(b_t)}$ occurring therein at any step $t$ in which a restart does not occur —the regret for the time steps when a restart occurs is already accounted for by the term $\mathcal{O}\big((\ln K)\ln(KT)\big)$ in (3). Now, Lemma 11 gives

$$|R_t| = \mathcal{O}\big(\alpha(G_t)\ln K\big)\,.$$

If $\gamma_t = \gamma_t^{(b_t)}$ for any time $t$ when a restart does not occur, it is not hard to see that $\gamma_t = \Omega\big(\sqrt{(\ln K)/(KT)}\big)$. Moreover, Lemma 13 states that

$$Q_t = \mathcal{O}\big(\alpha(G_t)\ln(K^2/\gamma_t) + |R_t|\big) = \mathcal{O}\big(\alpha(G_t)\ln(K/\gamma_t)\big)\,.$$

Hence,

$$Q_t = \mathcal{O}\big(\alpha(G_t)\ln(KT)\big).$$

Putting together as in (3) gives the desired result. $\qquad\square$