[Reviews · NeurIPS 2013]

Submitted by Assigned_Reviewer_1

The authors of this paper consider a kind of sequential prediction problem with a side graph information. It is a setting that is intermediary between the expert setting and the bandit setting, and where one observes, after choosing an arm, not only the outcome of the arm but also the outcome of some other arms that are linked to this arm through a graph. They consider two main settings, (i) the informed setting where the graph is made available to the learner before each prediction and (ii) the uniformed setting which they introduce and where the graph is only made available to the learner after his prediction. Their results also depend on whether the graph is directed or not. The results are provided in the adversarial case.

The authors provide some results in these two setting (uninformed and informed). They first provide an algorithm in the uniformed setting which I believe are the first ones in this setting, and bound the regret in the undirected and the directed case. In the directed case, there is a gap between the lower and upper bound for the first algorithm they introduced. In the informed setting, they manages, in the directed case, to go closer to the lower bound by building a new algorithm.

I find the paper well written and the results interesting. My main comment on it would be that it would gain in clarity if the authors could:
- write explicitly in the bandit and expert setting the bound of EXP3-SET, and comment on the fact that it matches the lower bound (up to some log terms in the bandit case) in both cases. The bound in the case of Erdos-Renyi random graphs is nice but gives less insight and you can put it in appendix if you lack space.
- To me Theorem 1 is a technical side results, and Corollary 2 is more clear. You should maybe delete Theorem 1 and keep corollary 2.
- I think it is very important that you extend a bit the explanation right after Corollary 2 on how to choose the \alpha_t, and of how much is the loss if you do not know them for tuning \beta. What you say makes sense, but it is too concise and have not a clear picture of how much you loose in the regret by doing what you propose.
- Maybe even, you could suppress Corollary 2 and just keep Corollary 4. And then, you will have more space to do an extended discussion.
- It would really help the reader to discuss the links and differences between Corollary 4 and Corollary 8 (again, to me, you can suppress Theorem 7 and just keep Corollary 8). For instance, provide an example of graph where \alpha(G) is small and mas(G) is large: after all, it is not obvious from your results that the bound in Corollary 8 is sharpen than the bound in Corollary 4 (even though the setting in Corollary 8 supposes that the graph information is available to the predictor), since you have an additional ln(K)\sqrt{\log(T)} that appears. Please discuss this in details.

As a conclusion I like this paper, and find the results nice. It seems correct but I did not have time to check the proofs in full details.
Summary: I find this paper very nice, and the results interesting and new. I did not check the proofs in the supplementary material though.

Submitted by Assigned_Reviewer_4

The paper extends the results of Shamir and Mannor ([11] in this paper). Shamir and Mannor studied interpolation between the bandit feedback and the full-info feedback in the problem of prediction with an expert advice. They consider the problem when, after the learner's prediction, the losses of several experts are revealed. Their main result shows that if an undirected graph G model the feedback (precisely, the nodes of G correspond to the experts and there is an edge ij iff after predicting i, the learner sees the loss of the expert j), then the optimal regret is \sqrt{T\alpha(G)}. Here, \alpha(G) is the independence number of G.

The main weakness of Shamir and Mannor is that the graph G is assumed to be undirected. The current paper gives upper bounds on the regret for the directed case. The author's bounds generalize the bounds of Shamir and Mannor in the sense that they (almost) coincide in the case that G is undirected. However, this paper's bound are not tight for every underlying graph G.

The current paper also strengthen the result of Shamir and Mannor in several other ways:
1. The presented algorithm is much faster.
2. In the case that the graph G changes in each round, the algorithm (as opposed to Shamir and Mannor) don't need to know G_t before its prediction y_t.

I find the results very interesting. The paper indeed strengthen Shamir and Mannor in several ways.
Summary: Very nice result. Strengthen Shamir and Mannor in several ways.

Submitted by Assigned_Reviewer_5

Summary:
--------
The authors provide improved, simplified, and extended analysis of adversarial multiarmed bandits (MAB) with additional observations that was introduced by Mannor and Shamir 2011. The actions in this game correspond to nodes in a graph and when playing an action the player observes the reward of the action played and of all the neighboring actions in the graph. The authors consider multiple settings of this game classified by (1) directed and undirected observation graphs, (2) adversarial or random graphs, and (3) informed and uninformed settings (in the informed setting the graph is observed before the action is played). For adversarial undirected uninformed case the authors provide matching upper and lower bounds (the lower bound is "inherited" from Mannor and Shamir), the algorithm is simpler and more efficient than the algorithm of Mannor and Shamir and generalizes the result of Mannor and Shamir to the uninformed case. The authors also provide algorithms for random (Erdos-Renui) directed uninformed case (without a lower bound), for uninformed directed adversarial case (with a certain gap to the lower bound) and for informed directed adversarial case (matching lower bound up to logarithmic terms).

Quality, clarity, originality, and significance:
------------------------------------------------
The paper is clear and of high quality. Concerning originality: the problem and tools for its analysis are known, however, there is still sufficient degree of originality in the combination. The paper is significant.

Additional comments:
--------------------
- The first motivation example sounds very similar to combinatorial bandits with semi-bandit feedback. Does it make sense to cast this problem as a bandit problem with observations graph? Does it give any advantage? How the results compare? Or, maybe, this is not a good example and better to be removed?

- You claim that you study random graphs, but in fact you study only one type of random graphs, the Erdos-Renyi model. I think this should be stated more explicitly.
Summary: The paper provides simplified, more efficient and general results for an important problem that was introduced at NIPS-2011. I think the paper will be interesting to the community.
Author Feedback

Author rebuttal: Reviewer 1:
Thanks for your excellent presentational suggestions. We will take them into serious consideration.


Reviewer 4:
It was not our intention to write the paper in a misleading way. We will rephrase the abstract to better clarify the match between upper and lower bounds in the various settings. More specifically, though there is some gap between Corollary 4 and Corollary 5 in the uninformed setting, this gap disappears (up to log factors) in the informed setting, just like in the paper of Mannor and Shamir which our lower bound argument builds upon. Please compare Corollary 5 to Corollary 8. Admittedly, we should have made this point clearer in the paper.


Reviewer 5:
Thanks for your detailed comments. As for your major comments:

- It is possible that the routing example may be viewed as an instance of combinatorial bandits. Our model is more general, since it does not assume linear losses, which in the routing example apparently arise from the partial ordering of sub-routes. We will explore this connection in the next version of the paper.

- We will better point out that our random graph analysis only refers to Erdos-Renyi
BTW: Lower bounds in this case can be proven, at least in the undirected case, by resorting to known results. See, e.g., A. Frieze, On the independence number of random graphs, Discrete Mathematics 81 (1990), 171–175, and refs. therein. Combined with Corollary 5, this essentially shows that, when r is constant, the inverse dependence on r is inevitable. We will point this out in the paper, thanks.